# Nanoscale kinetics of asymmetrical corrosion in core-shell nanoparticles

Hao Shan[1], Wenpei Gao[2], Yalin Xiong[3,4], Fenglei Shi[1], Yucong Yan[3], Yanling Ma[1], Wen Shang[1], Peng Tao[1], Chengyi Song[1], Tao Deng[1], Hui Zhang[3], Deren Yang[3], Xiaoqing Pan[2,5] & Jianbo Wu[1]

Designing new materials and structure to sustain the corrosion during operation requires better understanding on the corrosion dynamics. Observation on how the corrosion proceeds in atomic scale is thus critical. Here, using a liquid cell, we studied the real-time corrosion process of palladium@platinum (Pd@Pt) core-shell nanocubes via transmission electron microscopy (TEM). The results revealed that multiple etching pathways operatively contribute to the morphology evolution during corrosion, including galvanic etching on non-defected sites with slow kinetics and halogen-induced etching at defected sites at faster rates. Corners are the preferential corrosion sites; both etching pathways are mutually restricted during corrosion. Those insights on the interaction of nanostructures with reactive liquid environments can help better engineer the surface structure to improve the stability of electrocatalysts as well as design a new porous structure that may provide more active sites for catalysis.

[1] State Key Laboratory of Metal Matrix Composites, School of Materials Science and Engineering, Shanghai Jiao Tong University, 800 Dongchuan Rd, Shanghai 200240, People's Republic of China. [2] Department of Chemical Engineering and Materials Science, University of California, Irvine, Irvine, CA 92697, USA. [3] State Key Laboratory of Silicon Materials, School of Materials Science & Engineering, Zhejiang University, Hangzhou, Zhejiang 310027, People's Republic of China. [4] Hydrogen Energy R&D Department, Chemistry & Physics Center, National Institute of Clean-and-Low-Carbon Energy, Beijing 102211, People's Republic of China. [5] Department of Physics and Astronomy, University of California, Irvine, Irvine, CA 92697, USA. These authors contributed equally: Hao Shan, Wenpei Gao. Correspondence and requests for materials should be addressed to H.Z. (email: msezhanghui@zju.edu.cn) or to X.P. (email: xiaoqing.pan@uci.edu) or to J.W. (email: jianbowu@sjtu.edu.cn)

Platinum (Pt)-based nanoparticles continue to be the most widely used catalysts for oxygen reduction reaction (ORR) at the cathode of fuel cell, due to their potential advantages in both catalytic activity and stability[1–6]. However, the activity degradation arising from the loss of specific shapes and element dissolution remains an obstacle for widespread commercialization, despite the tremendous efforts devoted to enhancing the ORR properties of Pt-based nanoparticles through size-, shape-, and structure-control[7–11]. Recently, M-Pt (M = Pd, Au, Co, etc.) core-shell catalysts have proven to be one of the most promising systems that provide high activity, improved stability and efficient utilization of Pt[10,12–21]. The values based on liquid half cells have met the requirement for commercialization, but only 12~36% of the performance could be preserved when making into full fuel cell due to the use of different electrode, electrolyte, the different evaluation protocols and operating conditions[22–27]. The structure change during operation also leads to the deterioration of performance. It has been reported that the under−coordinated atoms on the surfaces can be protected by depositing or alloying with Au, adsorption of Br−, annealing and engineering the mesoporous structures[13,28–31]. However, the unavoidable loss of active metal by acidic corrosion during catalysis still restricts the practical application[32–35]. Therefore, it is urgent to understand the evolution of nanoparticles and the mechanism of nanoparticle-based corrosion with an aim to the long-term durable catalysts. On the other hand, recent effort in the design of active ORR electrocatalysts reveals that it is possible to employ the controlled dissolution of transition metal to obtain nanoframes, nanocages and jagged nanowires, which expose active Pt sites much more efficiently[4,10,36,37]. Exploration on the dynamics of the intermediate states is therefore the key to reveal the kinetics of not only the catalyst degradation, but also the formation of those highly active nanostructures; both are indispensable to the design of active and durable catalysts.

In situ techniques, including ICP-MS and Bragg coherent diffractive imaging (gBCDI), have been used to study the dissolution of metal electrodes in electrochemistry and morphology change of polycrystalline materials during operation[38–41]. However, structure evolution, including specific shape and morphology changes occurring locally on individual nanoparticles at the scale of nm can only be revealed using techniques with higher spatial resolution. To this end, in situ environmental liquid cell in transmission electron microscopy (TEM) has been demonstrated an effective way to study the real-time process of liquid-phase reactions, including the growth and dissolution of nanoparticles[42–48], some works even approached atomic resolution. While the chemical reactions studied in situ are fundamental in materials processes, the findings are not trivial, which are unknown or uncertain before the in situ observation. For example, the non-equilibrium states of nanostructures and their dissolution dynamics related to the local geometry can only be revealed by this means recently[35,48]. Especially, by analysis on the kinetics approaching atomic scale (sub-nm), shape anisotropy, which could be difficult to distinguish in static characterization before can now be derived, such as identifying the projected edge and corner sites of nanocubes[48]. The observations on the dynamics, therefore, not only reveal the structure-dependent kinetics, but can also impact the design of materials structures with more emphasis on the stable and/or active sites.

In this work, Pd@Pt core-shell nanocubes, which have been proven one of the active ORR catalysts[12,49,50], are employed as a model system to study the dynamic process of corrosion and nanocage formation by in situ liquid cell TEM. The results reveal that there are two corrosion pathways co-existing in the entire etching process, which are identified as halogen etching on exposed Pd surface and galvanic dissolution at the interface between Pt and Pd. Both etching mechanisms competitively contribute to the dynamic processes of corrosion of Pd and formation of Pt cages (see the details of the corrosion in Methods).

## Results

**In situ corrosion in Pd@Pt cubes.** We first revealed and compared the structure evolution of regular and corner defected Pd@Pt nanocubes to investigate the corrosion dynamics. Figure 1a shows the representative sequential TEM images of a regular Pd@Pt cube during the galvanic corrosion process. At the beginning, the Pd@Pt cube exhibited a well-defined cubic structure (Fig. 1a, 0 s). As etching proceeded, internal Pd atoms at the corner were etched first, leaving small voids at four corners (Fig. 1a, 7 s). Then, the four voids grew larger, at the meantime, the contrast of the four voids became much lighter, indicating that more Pd atoms were etched. However, the Pt shell was preserved. Afterwards, the Pd cubic core was etched from corners to center gradually until the entire interior part of Pd was etched away, leaving behind a Pt cage (Fig. 1a, 47–60 s). The Pt cage eventually shrunk into hollow sphere due to the difference in pressures between inside and outside (Supplementary Fig. 3 and Supplementary Movie 2)[10,51]. Overall, the etching started preferentially from corners, which may be ascribed to the high surface energy of corner site[52].

In comparison, defects at the corners of the particle can significantly change the etching scenario (Fig. 1b). At the defect, the Pd atoms exposed to the Br− electrolyte could be oxidized and dissolved in the form of [PdBr4]2−, leading to the further development of defects and the continuous etching of interior Pd (Fig. 1b). Small voids appeared at the four corners at 6 s, which is similar to the regular cubes. However, two voids at upper−left (UL) and lower−right (LR) corners grew faster, then coalesced into a nanochannel (Fig. 1b, 33 s). At 37 s, the defect at the UL corner expanded into a bigger void and the nanochannel grew bigger. Nonetheless, the other two corners rarely changed. Supplementary Fig. 4 and Supplementary Movie 3 show the entire etching process of this corner defected Pd@Pt cube. The etching rate of corner defected core-shell nanoparticle was faster than that of the regular one, which is due to the fast halogen etching in the defected areas[53]. We measured the sequential evolution of etching area ($C_a$) and etching rate ($C_r$) of both types of Pd@Pt cubes, respectively. From the schematics in Fig. 1c, the $C_a$ increased gradually along different directions in each cube. In the regular particle, the two curves of $C_a$ along UL–LR (solid orange arrow) and UR–LL (dashed orange arrow) were similar, indicating comparable etching behaviors during 1–47 s (Fig. 1d). The corresponding $C_r$ ($s^{-1}$), which is derived from the slopes of the fitted $C_a$, overlapped substantially (Fig. 1e). In the corner defected particle, the $C_a$ along UL–LR (with defects on the corners, solid green arrow) was much higher than that along UR–LL (dashed green arrow, Fig. 1d), which increased slowly and showed slight change after 10 s. Along UL–LR, $C_a$ followed an "S" curve. It raised quickly in 1–5 s, then slowed down in 6–23 s, and finally increased rapidly again in 24–32 s. The $C_r$ along UL–LR was higher than that along UR–LL during most of the etching process (Fig. 1e) because the etching was promoted by the Br− ions induced corrosion with relatively lower redox potential, in comparison to galvanic etching at other corners without defects. It is worthwhile to note that along UR–LL, where the corners were defect free in both types of particles, the values of $C_r$ are still different. Indeed, $C_r$ along UR–LL in the defected cube was lower than that of the regular cube throughout the entire process because galvanic corrosion on these corners would be inhibited due to the overconsumption of electrons from halogen etching at the defected corner of the same cube.

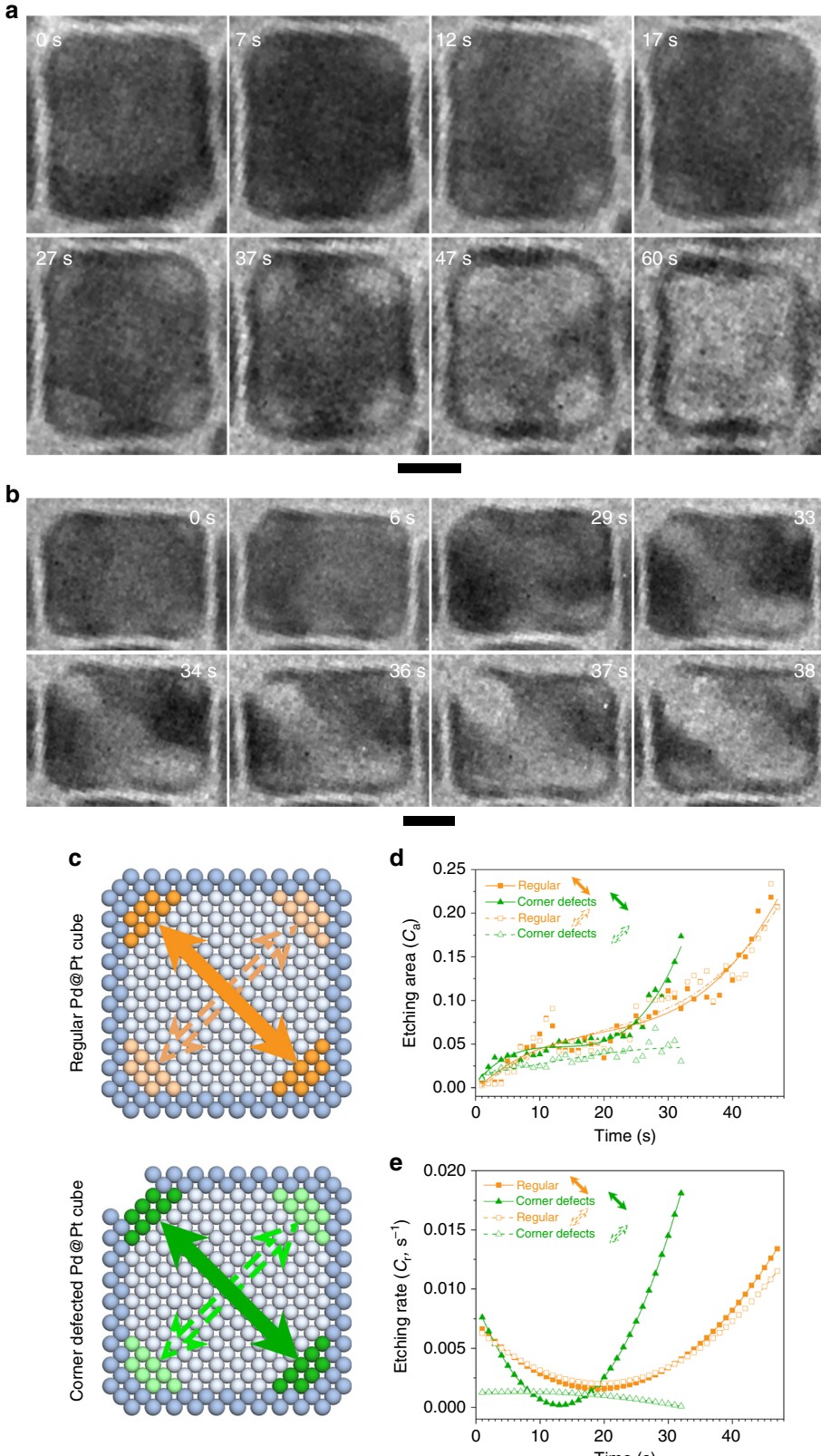

**Fig. 1** Etching process of regular and corner defected cubes. **a**, **b** Time sequential TEM micrographs showing the etching process of internal Pd atoms in a single regular and corner defected Pd@Pt cube, respectively. Scale bars in all panels are 5 nm. **c** The illustration of atomic structures demonstrates the calculation of $C_a$. **d** Scatter diagrams and fitting curves of $C_a$. **e** Corresponding rates of etching areas in **d**, of each direction in two types of cubes

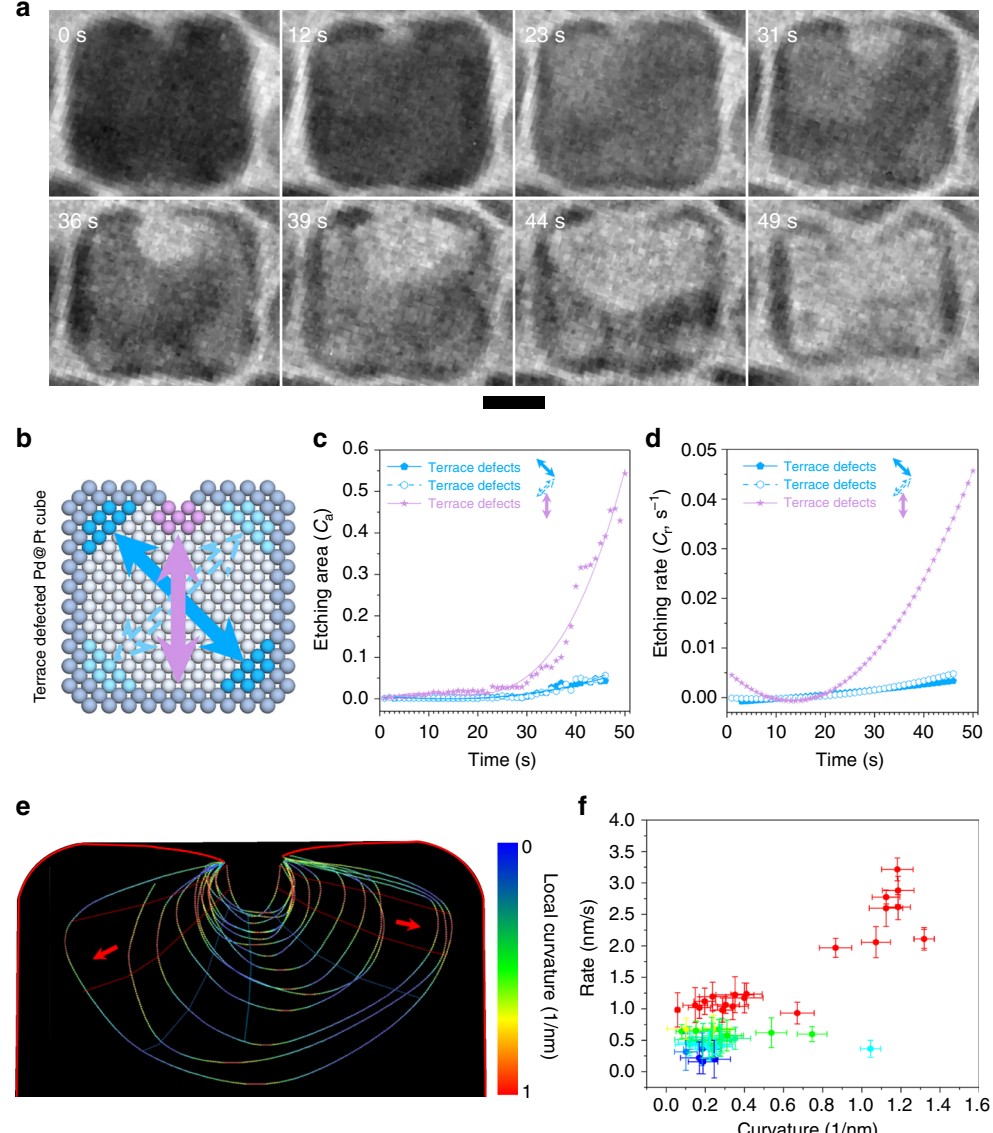

**Fig. 2** Etching process of terrace defected cube. **a** Time sequential TEM micrographs showing the etching process of internal Pd atoms in a single terrace defected Pd@Pt cube. Scale bar is 5 nm. **b** The illustration of atomic structures demonstrates the calculation of $C_a$. **c** Scatter diagrams and fitting curves of $C_a$. **d** Corresponding rates of etching areas in **c**, along each direction in the cube. **e** Time-domain contour plots of etched hole along the direction of terrace defects. Contour lines are spaced in time by 2 s. Color of curves shows the local curvature. **f** Relationship of the etching rate and local curvature. The measuring dimensions (lengths) have the error of ±1 pixel in the image, the error bars are then calculated following error analysis during the derivation of curvature and etching rate

For the cubic nanoparticle with defects on terrace, no void nucleated at the corners except for the cavity at terrace during the corrosion process (Fig. 2a, 0 s). Supplementary Fig. 5 and Supplementary Movie 4 show the entire etching process of Pd@Pt cubes with defects on terrace. The cavity nucleated, then grew deeper and bigger at 12 s, while only small pinholes appeared at the four corners at this moment. Since then, the cavity expanded to the two sides and swallowed the holes in UR and UL corners at 39 s and 44 s, respectively. No obvious growth of the pinholes was observed on the two bottom corners. Afterwards, the residual Pd atoms in the nanostructure were dissolved rapidly within 5 s, leaving a hollow Pt cage. The different intermediate states of Pd@Pt cube with terrace defects were also observed in the STEM images in Supplementary Fig. 6. The plots in Fig. 2b–d show the corrosion kinetics of terrace defected cube along three directions. Each $C_a$ increased gradually (Fig. 2c). Among them, $C_a$ originated from the defect on terrace

(solid purple arrow) changed slightly in the first 29 s, and then increased dramatically in 30–50 s, while $C_a$ along UL–LR (solid blue arrow) and UR–LL (dashed blue arrow) increased slowly, which was similar to that of the defect-free corner in the corner defected cube. Besides, Fig. 2d demonstrated that $C_r$ along UL–LR and UR–LL changed slightly in the entire etching process.

During the development of large cavity originated from the defect on the terrace, we find the cavity evolution via halogen etching highly depends on the local geometry. Figure 2e shows the contour evolution of the etched cavity with the colored curvature (see the details of how to establish the contour with curvature in Methods section). The two horizontal sides of the contours featured with large curvature were generally etched faster, while other areas with small curvature show relatively smooth transition, which reveals that this surface etching highly relied on the local curvature. However, the evolution of these two left and right regions are not the same because of the variation of

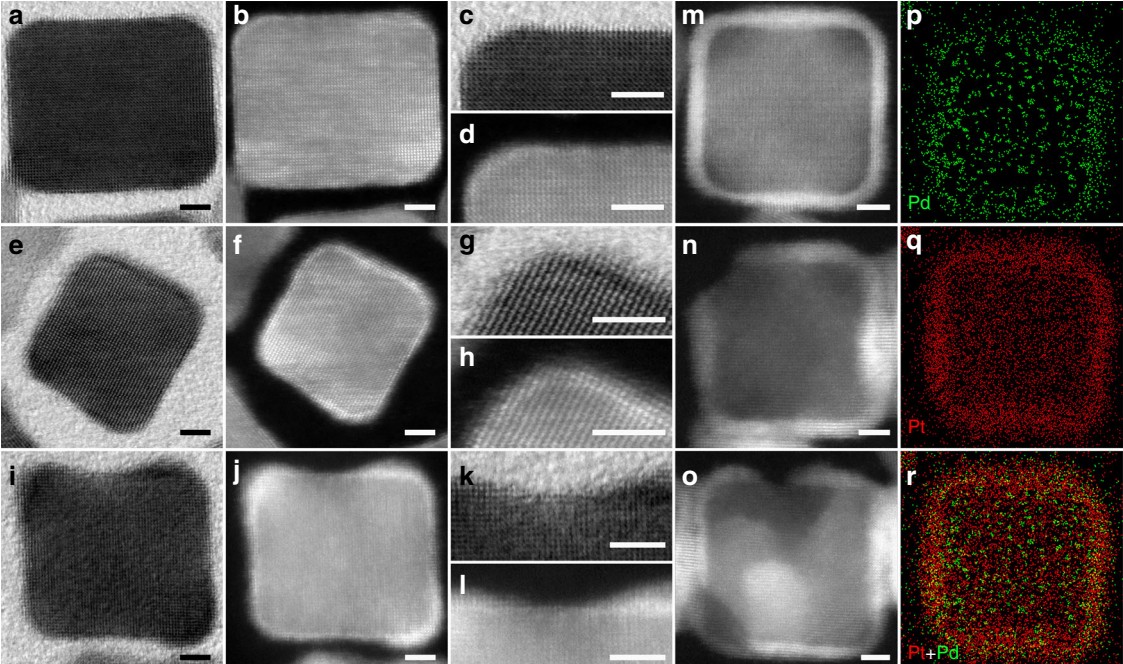

**Fig. 3** HRTEM and HAADF-STEM characterizations of cubes. **a–l** HRTEM and HAADF-STEM micrographs of regular (**a–d**), corner (**e–h**) and terrace (**i–l**) defected Pd@Pt cubes before etching. **m–o** Pt cages derived from cubes of regular, with corner and terrace defects, respectively. **p–r** EDS elemental mapping of Pt cage in **m**. Scale bars in all panels are 2 nm

Br⁻ ions density. Figure 2f shows a relationship of the etching rate and local curvature, in which the highly curved region proceeded fast at $3.5\ nm\ s^{-1}$ while the region with a curvature of $0.2\ nm^{-1}$ was etched at $0.5\ nm\ s^{-1}$. This relationship is because the highly curved region usually has low coordinated sites with high surface energy. Such loosely bonded surface atoms can relax locally by surface diffusion[35,52,54], and therefore can react with the etching electrolytes at a higher rate. The formation and evolution of the cavity, therefore, was driven by the preferential etching along the direction perpendicular to the highly curved area as indicated by the red arrows (Fig. 2e).

**Structural and compositional characterization**. High-resolution transmission electron microscopy (HRTEM) and high-angle annular dark-field scanning transmission electron microscopy (HAADF-STEM) micrographs reveal the detailed structures and compositions of the samples in different stages of etching (Fig. 3). The as-synthesized Pd@Pt cubes mostly have ultrathin Pt shells with 3–4 atomic layers deposited on the surface of Pd cube uniformly (Fig. 3a–d)[12]. However, some of the Pd@Pt cubes were not fully covered by Pt, showing some defects on corner and (100) terrace, respectively (Fig. 3e–l). Throughout the etching processes, Pt cages were obtained after internal Pd atoms were etched away, as shown in Fig. 3m–o. The defects on corner and terrace still existed (Fig. 3n, o). Energy dispersive spectra (EDS) elemental mapping of the regular cube (Fig. 3p–r) reveals that compared to the Pd@Pt cube before etching (Supplementary Fig. 7), only trace Pd was found in the inner region of Pt cage after etching, indicating most of internal Pd atoms have been etched. A large amount of Pt cages were shown in Supplementary Fig. 8, which indicated that the etching process reliably occurred in most of Pd@Pt cubes. In addition, some Pt cages with defects collapsed into hollow spheres in the end (Supplementary Fig. 9).

It has been reported that Pt element can be dissolved and rearranged during electrochemical potential cycling, where Pt can be oxidized into Pt-O and reduced[36,55], and the inner part of active metal, such as Pd may be dissolved as well[12,19]. These processes result in the change on the particle structure. In our case, the process of halide-induced oxidative etching is similar to the oxidative etching in electrochemistry. Only that the halide-etching is more significant than the formation of metal-oxide, considering that the oxidation of Pd in the form of $[PdBr_4]^{2-}$ could be thermodynamically easier than the formation of oxides due to the redox potential difference between the two reactions (Eqs. 1 and 3). On bimetallic nanoparticles, while halide-induced corrosion of Pt was also possible, where Pt atoms on the shell could be etched by Br⁻ to form $[PtBr_4]^{2-}$, this process was largely suppressed in the presence of Pd, which serves as the anode in our case during galvanic corrosion. However, Pt atoms might still be etched due to the loss of the Pd support underneath. In the course of oxidative etching, the inter⁻diffusion between Pd and Pt might also occur, which facilitated the migration of Pd atoms from inside into Pt shell and resulted in the alloying of Pd–Pt in the shell, according to previous reports[10,31,56–60].

**Site dependence on corrosion rate**. We selected three scatter diagrams of representative $C_a$ as a function of time for the three types of cubic nanoparticles (Fig. 4a). The average etching rate within different periods of time for each cube in the inset table (also in Supplementary Table 1) shows an order of corrosion: Stage III > Stage I > Stage II. Indeed, corrosion that occurs on corner follows this "S" type three-stage process, which is highly related to the coordination number (CN) of the atoms exposed, and therefore the evolution of surface geometry[35]. The etching favors its initialization from corners (Stage I), which has the lowest CN of 3 (Fig. 4b)[61]. This resulted in the exposure of {111} and {110} surface facets gradually, on which CN was 5–7 and 6–7, respectively. The appearance of these two more stable facets resulted in the retarded etching of Stage II. In the end, after these low-index atoms were etched away, quasi-spheres with high index planes remained, experiencing a rapid etching, labeled as Stage III[62]. However, in the case of the terrace defected cube, only halogen etching occurred at terrace, at which CN was 8, resulting in the slowest etching rate on terrace {100} (Supplementary

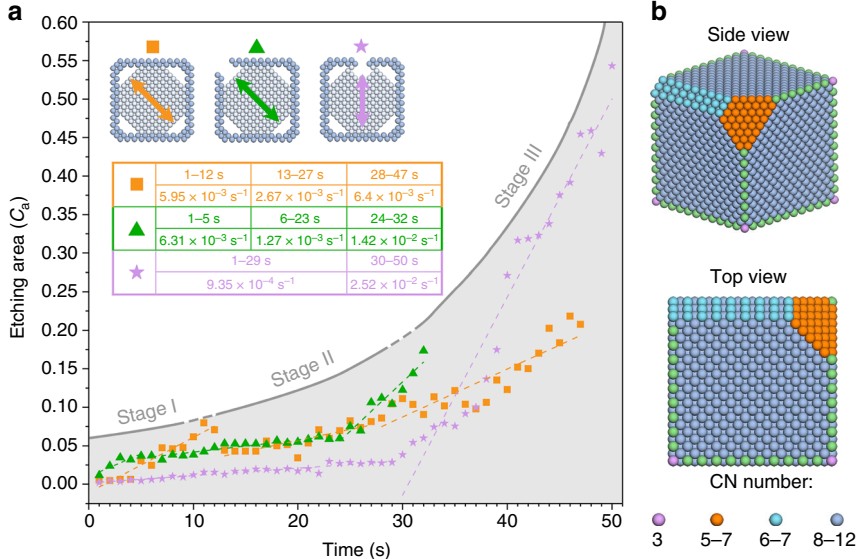

**Fig. 4** Etching rate and coordination number of cubes. **a** Changes of the measured projected etching area along representative directions as a function of time in regular, corner and terrace defected cubes, respectively. Inset table shows the values of average etching rate in the corresponding periods of time. **b** Coordination number distribution of surface atoms of inside Pd cube

Note 1 and Supplementary Figs. 11–13). Then, halogen etching was promoted at the highly curved sites, where the atoms also bear low CN, leading to the preferential etching direction towards the high curvature. Similar behavior was also seen in the formation of nanochannel in the corner defected cube, the tips of nanochannels along the etching direction are always highly curved. In defected cubes, galvanic etching and halogen etching start at the same time: we observed that the former one was inhibited by the latter one. The faster the halogen etching was, the more the galvanic etching was suppressed. The competition between these two corrosion pathways resulted in that the galvanic etching at the corner of terrace defected cube is faster than that of corner defected cube. The order of the etching rate of galvanic etching is regular cube > terrace defected cube > corner defected cube (Supplementary Note 2 and Supplementary Figs. 14–16).

Now we know that within one Pd@Pt cube, the halogen etching only occurred at the defected sites, where Pd atoms were exposed to Br$^-$ ions directly, while the galvanic etching was preferentially initiated at corners rather than anywhere else. In defected cube, a competition existed between galvanic etching and halogen etching. When we summed the $C_a$ ($\Sigma C_a$) over the entire cube as a function of time, which is an overall corrosion from both galvanic and halogen etching in Supplementary Fig. 17, we surprisingly found that the curves of $\Sigma C_a$ coincided for cubes without defects and with defect on the corner during the first 30 s. Namely, the amounts of Pd atoms etched was almost the same for these two types of cubes at the early stage when the stable structures still had distinctive {111} small facets on the corners, regardless of the etching mode. However, after the Pd cubic core lost the cubic morphology, halogen etching became faster in corner$^-$ defected cube. The cube with terrace defect experienced a slower and longer etching process, which indicates that corner is the favorable etching site. Supplementary Movie 5 shows the entire etching process of the three types of Pd@Pt cube. Thus, the protection or coating of corner sites is primary for designing durable Pd@Pt electrocatalysts in the long term.

In summary, we have captured the etching process of Pd@Pt cubes and the formation of Pt cages in real-time in a liquid cell TEM, which identified two types of etching pathways, including slow indirect-contacting galvanic etching on non-defected site

and fast direct-contacting halogen-induced etching at defected site, contributing to the corrosion and morphology evolution of nanostructures. The halogen etching and the galvanic etching are simultaneously operative and competing because of the limited consumption of electrons in the reaction. The two etching pathways both have the geometry preferences that the corner is the initial and faster etching site, which reveals the important role of protecting corners of electrocatalyst to improve the applicable stability. These corrosion kinetics here revealed at the nanoscale provide insights towards engineering of the surface defect to pursue multimetallic core-shell electrocatalysts with improved stability for ORR in fuel cell. The etching process, if under control, could also help to design more stable catalysts with porous or hollow architectures, which may provide more active sites in catalysis[63]. It is worth noticing that correlative study using various in situ approaches can offer more comprehensive understanding on the structure evolution and mechanisms. Such study can largely impact emerging researches not only in fuel cell, but also on other electrochemical reactions involved in renewable energy for replicable petrochemical fuel, zero-emission of carbon and hazard water treatment such as water splitting, CO$_2$ reduction, and peroxide generation.

## Methods

**Chemicals and materials**. Sodium tetrachloropalladate(II) (Na$_2$PdCl$_4$, 99.99%), poly(vinylpyrrolidone) (PVP, Mw ≈ 55,000), ascorbic acid (AA), potassium bromide (KBr) and chloroplatinic(IV) acid (H$_2$PtCl$_6$, 99.9%) were purchased from Sigma-Aldrich. Oleylamine (OAm, 80–90%) was purchased from Aladdin. Toluene, ethanol, cyclohexane, acetone were purchased from Sinopharm Chemical Reagent Co., Ltd. Deionized water (DI water) used was 18.2 MΩ cm. All the chemicals and materials were used as received.

**Synthesis of Pd@Pt core-shell nanocubes**. The Pd cubic seeds with an average edge length of ~ 12 nm were prepared according to the previously reported method[64]. The as-synthesized Pd nanocubes were dispersed into OAm by a phase transfer method. After that, the ultrathin Pt layers were epitaxially grown on the surface of Pd nanocubes by a seed-mediated method[12].

**Synthesis of Pd nanocubes**. In a typical synthesis, 105 mg of PVP, 60 mg of AA, 400 mg of KBr, and 8.0 mL of DI water were mixed in a 20 mL glass vial and the solution was preheated to 80 °C for 10 min under magnetic stirring. After that, 3 mL of DI water containing 57 mg of Na$_2$PdCl$_4$ was quickly injected into the preheated solution by a pipette. The vial was capped and the reaction was kept at 80 °C

for 3 h. After the mixture had cooled to room temperature, the product was obtained by centrifugation, washed with ethanol and acetone for three times, and then re-dispersed in ethanol for further use.

**Phase transfer of Pd cubes**. In a standard procedure, 8 mL of ethanol solution containing the Pd cubes were mixed with 5 mL of OAm and 3 mL of toluene in a 20 mL glass vial. The phase transfer was conducted by magnetic stirring of the mixture at 80 °C for 9 h. The product was collected by centrifugation, washed with ethanol, and then re-dispersed in OAm serving as the seeds for the epitaxial growth of ultrathin Pt layers.

**Epitaxial growth of ultrathin Pt layers on Pd cubes**. In a standard preparation, 4 mL of OAm solution containing cubic Pd seeds were added into a 25 mL single-necked flask and preheated to 180 °C in an oil bath for 10 min. After that, 2 mL of OAm solution containing a certain amount of $H_2PtCl_6$ was added into the flask by a pipette. The reaction was proceeded at 180 °C for 3 h. Finally, the final product was collected by centrifugation, washed with ethanol and cyclohexane for several times.

**Materials characterization**. High-resolution transmission electron microscopy (HRTEM) and high-angle annular dark-field scanning TEM (HAADF-STEM) and energy dispersive X-ray (EDX) mapping analyses were taken on JEM-ARM200F with a cold field emission gun and a spherical aberration corrector equipped with an EDX detector system of EX-230 BU_37001-2 produced by JEOL. The model of the aberration corrector was CEOS GmbH.

**In situ TEM characterizations**. We used the liquid flow TEM holder Poseidon 500 (Protochips, North Carolina, USA). A pair of chips were used. Two thin silicon nitride (SiN) membranes of the liquid cell window (50 nm thick for each membrane, $550 \times 20$ μm for window) encapsulatesthe gap of 50 nm, allowing reaction liquid (0.1 M NaBr) flowing at a flow rate of 5 μl/min. Prior to loading, photoresist of two chips needed to be removed by immersing them into acetone for 2 min, and then wet transfer to methanol for 2 min. After drying by compressed air, the chips were placed on a glass slide and then sent into oxygen plasma cleaner (Gatan, Model 950) for 5 min in order to remove organic contamination. The small chip was on the tip of the in situ TEM holder and 1–2 μl of solution was dropped onto it with a pipette, the small chip was covered by the large chip eventually. Leak checking was performed in a home-made vacuum pump before inserting the in situ holder into TEM. The in situ TEM imaging was carried out on a JEOL JEM2100 (JEOL, Tokyo, Japan) with a $LaB_6$ emitter at 200 kV. The liquid flow rate was 5 μl/min controlled by syringe pump (Harvard Apparatus, Pump 11 Elite) (Supplementary Movie 1). Videos were recorded using the Cantega G2 camera (Olympus, Tokyo, Japan). Electron beam current density used in the experiment was 68 pA/cm$^2$. All image analysis was done on the original images extracted from the recorded videos. We used ImageJ software and DigitalMicrograph software to measure the distance and area in etching, respectively.

**Description of halogen etching and galvanic etching**. It is well known that the standard reduction potential of $Pd^{2+}/Pd$ pair is 0.92 V (vs. the standard hydrogen electrode, SHE), which is lower than that of $Pt^{2+}/Pt$ (1.2 V vs. SHE)[65]. On the basis of this difference in redox potential, Pd and Pt, may form galvanic cells in the presence of electrolyte[66], in which, Pd serves as the anode and Pt as the cathode. Within the cell, the current flows from Pd to Pt, and to the electrolyte, which promotes the etching of Pd and protects Pt from corrosion. Pt cages can be obtained from Pd@Pt cubes using such selective etching of Pd. The relevant galvanic etching reactions are as follows[67]:

$$Pd^{2+} + 2e^- \rightarrow Pd \; E = 0.92V \text{ vs. SHE} \quad (1)$$

$$O_2 + 4H^+ + 4e^- \rightarrow 2H_2O \; E = 1.23V \text{ vs. SHE} \quad (2)$$

In our in situ experiment using liquid cell, $O_2$ can be generated by beam irradiation[68,69]. When $Br^-$ ions are introduced, $Br^-$-induced oxidative etching can also occur along with galvanic etching[7,68,70], in which Pd atoms react with $Br^-$ and form $[PdBr_4]^{2-}$ with a standard reduction potential of 0.49 V. This halogen-induced oxidative etching is thus thermodynamically preferable than galvanic etching[71].The relevant reaction equation is as follows[67]:

$$[PdBr_4^{2-}] + 2e^- \rightarrow Pd + 4Br^- \; E = 0.49V \text{ vs. SHE.} \quad (3)$$

This resulted in the exposure of internal Pd atoms to the electrolyte directly and enabled the reaction of Pd atoms with $Br^-$ ions. Two types of defects resulted in different etching phenomena (Figs. 1b and 2a). When these defected Pd@Pt cubes encountered with solution, exposed Pd atoms at the defects sites (corner and terrace) processed a different halogen etching in the electrolyte containing $Br^-$ ions to form $[PdBr_4]^{2-}$ via Eq. 3[68]. However, at those sites without defects, as well as the surface sites in the regular cubes, $Br^-$ cannot directly contact the internal Pd atoms, normal galvanic etching would happen followed by Eq. 1, as described in Fig. 1a.

In addition, Pt atoms can also react with $Br^-$ to form $[PtBr_4]^{2-}$ with a standard reduction potential of 0.698 V, which is larger than $[PdBr_4]^{2-}$. The relevant equation is[67]:

$$[PtBr_4^{2-}] + 2e^- \rightarrow Pt + 4Br^- \; E = 0.698V \text{ vs. SHE.} \quad (4)$$

It indicates that there would be Pt dissolution in the process, yet the amount of Pt dissolved is less than Pd.

**Calculation of etching area**. The calculation method is done using ImageJ software and shown in Supplementary Fig. 1. In a etched cube, the total etched area along the same direction divided by the whole area of cube is defined as etching area ($C_a$). Therefore, $C_a$ along the direction of UL–LR is (Area 1 + Area 4) / Area$_{cube}$ and $C_a$ along the direction of UR–LL is (Area 2 + Area 3) / Area$_{cube}$. The data are collected before two random holes are merged.

**Etched volume and etched atoms number**. The volume and number of atoms being etched were also calculated and analyzed based on the data of etched area measured aforehand. Because the TEM images are the projection of the 3D structure in 2D, it is challenging to estimate the etching volume based on the observed image, especially in such a highly dynamic nature; however, we can approach such estimation by assuming that the etched space is pseudo-spherical. This assumption is based on: (1) the etched areas usually exhibit a rounding exterior contour; (2) we always observe that during the enlarging of the etched area, the contrast from that area becomes brighter, indicating that the etching is also proceeding in the depth direction along the electron beam. Therefore, the etched volume is approximated as follows:

$$V_{etch} = \frac{4}{3}\pi r^3. \quad (5)$$

$$\text{Since}: A_{etch} = \pi r^2 \quad (6)$$

$$V_{etch} = \frac{4}{3}\pi \left(\frac{A_{etch}}{\pi}\right)^{\frac{3}{2}}. \quad (7)$$

Where the number of etched atoms was approximated as follows:

$$N = \frac{V_{etch} \times \rho_{Pd}}{M_{Pd}} \times N_A \quad (8)$$

Curves of the entire etched volume and etched atom number as a function of time were shown in Supplementary Fig. 2.

**Measurement of distance**. The distance of residual Pd atoms ($D_r$) along different directions was measured by ImageJ software and shown in Supplementary Fig. 10. Three directions were measured: upper-left to lower-right (UL–LR), upper-right to lower-left (UR–LL), and terrace defects direction, as shown in the cube with corner and terrace defects. Along each direction, only residual unetched Pd atoms are included. Distance was measured before a random direction was etched completely.

**Mapping the contours of cavity**. We outlined the edge of the newly formed cavity by sampling on the distinguishing border between the dark and bright contrast (the contours in Fig. 2b) and calculated the curvature of the edge, as shown by the color code of the etching contours.

It is worth noting that such preferential etching at highly curved area results in two controversial mode depending on the location. If etching occurs outside a nanostructure, those highly curved features will be etched first, the resulting structure will be getting close to isotropic spheroid, with smooth surface. However, if etching occurs inside the nanostructure, as the highly curved area are kept being consumed faster, even higher anisotropic interior structure forms.

**Data availability**. The data that support the findings of this study are available from the corresponding author on reasonable request.

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

## Acknowledgements

The work is sponsored by the thousand talents program for distinguished young scholars from Chinese government, National Key R&D Program of China (No. 2017YFB0406000) and the National Science Foundation of China (51521004 and 51420105009), and start-up fund (J.W.) and the Zhi-Yuan Endowed fund (T.D.) from Shanghai Jiao Tong University. The work at University of California-Irvine is supported by the National Science Foundation (NSF) under grant numbers CBET 1159240, DMR-1420620, and DMR-1506535. H.Z. acknowledged financial support by the National Science Foundation of China (51372222 and 51522103) and National Program for Support of Top-notch Young Professionals.

## Author contributions

H.S., T.D. and J.W. conceived and designed the study. Y.X. prepared samples. H.S. performed the in situ experiment. H.S., Y.X., Y.Y. and Y.M. carried out the HRTEM and HAADF-STEM characterization. H.S., W.G. and F.S. carried out the data analysis. W.S., P.T., C.S. and D.Y. helped discuss the results. H.S. and W.G. drew pictures; H.S., W.G., H.Z., and J.W. wrote the manuscript. H.Z., X.P., T.D. and J.W. managed the project and reviewed the results, data analysis, and manuscript preparation.

## Additional information

**Competing interests:** The authors declare no competing interests.

