## [Peer Review File · Nature Communications]

Reviewers' comments:

Reviewer #1 (Remarks to the Author):

This paper reports a very well executed insitu study of core shell nanoparticles of Pt and Pd that are used for ORR.

these experiments are not trivial and have been well done.

the authors have studied three types of particles with different defect sites and assessed their corrosion.

This paper will be important to a very broad audience including fuel cell chemists, nanoparticles synthesis and TEM experts and is highly suitable for Nature Comm.

My main comment for improvement is

From the equation in the supporting information the rates are calculated from changes in area observed in the TEM. However the nanoparticles are 3 dimensional not 2 dimensional objects.

An effort to calculate rate changes in term of volumes and numbers of atoms etched would be more meaningful.

Reviewer #2 (Remarks to the Author):

This is an excellent paper that delves into corrosion mechanisms and their relative importance and strength at different face of catalyst surfaces/edges, etc., It is a fundamental work of great import to PEMFCs. Such catalyst are being scaled up and applied to MEAs of fuel cells and don't work very well. This work throws light on the corrosion processes that may at some time in the future help mitigate the losses due to dissolution of catalyst.

There is one statement at the beginning of the paper that justifies the work stating "The values based on liquid half cells have met the requirement for 49 commercialization, but only 12~36% performance could be preserved when making full 50 fuel cell because of the unavoidable loss of active metal by acidic corrosion, which still 51 restricts the practical application²¹⁻²⁴"

This statement is incorrect to the extent that the ORR activities found in RDE and liquid half cells cannot be attained in fuel cells for many obvious reasons. The electrolyte and measurement method in RDE renders it useful only as a relative indicator of ORR activity rather than an absolute value. ORR activities of 15-20 times that of plain Pt/C have been obtained in RDE and these only give about 2-4 times Pt/C values in MEAs of fuel cells. For references please refer to RDE papers by Shinozaki et al and others from DOE over the last 2 years on benchmarking and best practices for RDE measurements. It has nothing to do with corrosion.

However, this work is still useful and important and may be applied to practical catalysts at least to explain their behavior.

Reviewer #3 (Remarks to the Author):

Report for MS by Shan et al.

The authors used an experimental and computational approach to study relationships between the coordination of surface atoms (shape of NPs) and dealloying-induced morphological changes of the Pt-Pd core shell structure. Although in situ TEM imaging is an important step in visualizing dealloying-induced transformations, this paper will only be suitable for publication in Nature Communications after a major revision. The authors must address the shortcomings listed below:

The abstract is misleading. This paper has nothing to do with understanding how to minimize the corrosion of ORR catalysts. It is, in fact, a study related to monitoring morphological changes during the dealloying of Pt/Pd systems; and this should clearly be stated in the paper. The conclusion that we have to learn how to protect unstable, low coordinated atoms is not new and has been comprehensively discussed in the literature that, unfortunately, the authors omitted to cite.

As mentioned above, it is important that it is now possible to use TEM for monitoring in situ morphological changes during dealloying of bimetallic systems. However, it is of paramount importance to use additional experimental probes that can provide atomic-scale information in order to be able to fully understand how the kinetics of dissolution depends on the density and nature of defects, the electrode potential, the electrochemical environment, and the history of the experiment. For example, in situ ICP-MS methods have been developed for studying single crystals, thin films and NPs, and in situ coherent X-ray diffraction has also been routinely used to study the role of defects in NPs. For details see: *Angew. Chemie Int. Ed.* 2012, 51 (50), 12613–12615; *Electrochem. Commun.* 48, 81–85 (2014); *ACS Catalysis*, 6 (2016) 2536-2544 and *Science* 356 (2017)739-742. Without these additional experimental techniques it is very easy to over-interpret the TEM results as, in my opinion, the authors have done in this manuscript.

The authors have no proof for the proposed mechanism of Pt/Pd dealloying. In particular, there is no proof that Pd dissolution is accompanied with the ORR. Currently, it is well established that the dissolution of Pt-based metals in halide-free solutions is triggered by an irreversible, potential-induced oxide formation/reduction process. In the presence of halides, however, the dissolution is governed by the competition between metal-oxide formation and metal-halide complexation. It is surprising that the authors did not cite any work dealing with such issues. It is also surprising that the authors omitted to cite a classical work of the proposed dealloying mechanism for bimetallic systems (e.g., *Nature* 410, 450–453 (2001), *J. Am. Chem. Soc.* 134, 8633–8645 (2012), and *Nat. Mater.* 9, 904–7 (2010)).

The authors proposed that only Pd dissolution takes place during the dealloying; why not Pt as well? Certainly, the kinetics of dissolution are much higher for Pd than Pt, but still a small amount of Pt dissolution may have a substantial effect on the formation of the “hollow structure”. The authors must provide evidence that only Pd dissolution takes place during the dealloying. The authors must also discuss the role of possible surface diffusion of the nobler element (Pt in this case) and how this may affect the final dealloyed structure.

In conclusion, the paper has nothing to do with understanding how to make more durable catalysts for the ORR, and this should clearly be stated in the manuscript. Dealloying is a synthesis method that has been used for decades to make new porous architectures that may provide more active sites for the reaction of interest. A new direction should be to learn how such porous materials may affect the stability of surface atoms during electrocatalytic processes. In the revised version of the paper the

authors must find a way to address correlations between their porous structure and the stability of this material.

Reviewer #1 (Remarks to the Author):

Comments: *“This paper reports a very well executed in situ study of core shell nanoparticles of Pt and Pd that are used for ORR.*

these experiments are not trivial and have been well done.

the authors have studied three types of particles with different defect sites and assessed their corrosion.

This paper will be important to a very broad audience including fuel cell chemists, nanoparticles synthesis and TEM experts and is highly suitable for Nature Comm.”

Response: We would like to thank the reviewer for the appreciation of our effort on the study of etching of core-shell nanoparticles using liquid cell. The results contribute to our knowledge on: the formation of Pt nanoframe; the etching of the less-expensive metal; and the competition of different etching modes. To improve the presentation of our study, we did further analysis on our data based on the reviewer’s suggestion.

Comments: *“My main comment for improvement is*

From the equation in the supporting information the rates are calculated from changes in area observed in the TEM. However the nanoparticles are 3 dimensional not 2 dimensional objects.

An effort to calculate rate changes in term of volumes and numbers of atoms etched would be more meaningful.”

Response: We agree with the reviewer on this point. Because the TEM images are the projection of the 3D structure in 2D, it is challenging to estimate the etching volume based on the observed image, especially in such a highly dynamic nature, however, we can approach such estimation by assuming that the etched space is pseudo-spherical. This assumption is based on that: 1) the etched areas usually exhibit a rounding exterior contour; 2) we always observe that during the enlarging of the etched area, the contrast from that area becomes brighter, indicating that the etching is also proceeding in the depth direction along the electron beam. Therefore the etched volume is approximated as:

$$V_{etch} = \frac{4}{3}\pi r^3$$
$$\text{Since: } A_{etch} = \pi r^2$$
$$V_{etch} = \frac{4}{3}\pi \left(\frac{A_{etch}}{\pi}\right)^{\frac{3}{2}}$$

The number of atoms being dissolved during the etching process is then:

$$N = \frac{V_{etch} * \rho_{Pd}}{M_{Pd}} * N_A$$

All the related calculation and analysis are now added in the Supplementary Information in the highlighted section of newly-added Supplementary Section 4 and Figure S2.

Reviewer #2 (Remarks to the Author):

Comments: *“This is an excellent paper that delves into corrosion mechanisms and their relative importance and strength at different facets of catalyst surfaces/edges, etc., It is a fundamental work of great importance to PEMFCs. Such catalyst are being scaled up and applied to MEAs of fuel cells and don’t work very well. This work throws light on the corrosion processes that may at some time in the future help mitigate the losses due to dissolution of catalyst.”*

Response: We thank the reviewer for acknowledging the importance of our work. The corrosion process studied here, we believe, is related to the potential failure of application of shape and composition controlled nanoparticles when being scaled up.

Comments: *“There is one statement at the beginning of the paper that justifies the work stating “The values based on liquid half cells have met the requirement for 49 commercialization, but only 12~36% performance could be preserved when making full 50 fuel cell because of the unavoidable loss of active metal by acidic corrosion, which still 51 restricts the practical application 21-24”*

This statement is incorrect to the extent that the ORR activities found in RDE and liquid half cells cannot be attained in fuel cells for many obvious reasons. The electrolyte and measurement method in RDE renders it useful only as a relative indicator of ORR activity rather than an absolute value. ORR activities of 15-20 times that of plain Pt/C have been obtained in RDE and these only give about 2-4 times Pt/C values in MEAs of fuel cells. For references please refer to RDE papers by Shinozaki et al and others from DOE over the last 2 years on benchmarking and best practices for RDE measurements. It has nothing to do with corrosion.”

However, this work is still useful and important and may be applied to practical catalysts at least to explain their behavior.

Response: Thanks for the referee’s comment. We have read and cited Shinozaki’s papers on the differences in RDE and MEAs, and corrected the description in the revised manuscript (see Line 49-51). We clarified that the poor performance of ORR catalyst in full fuel cell in comparison to ORR test is due to the use of different electrode, electrolyte, and the different evaluation protocols and operating conditions, specific to the following aspects:

1. The evaluation protocols are vastly different between RDE and MEA. Particularly, in RDE, the potential range from which kinetic information for the ORR can be obtained is narrow (>0.7-0.8 V vs. RHE).
2. The catalyst layer interface are different between two test conditions. In MEAs, the catalyst layer interface is porosity and ionomer coverage, while that in RDE is all acidic electrolyte.
3. Operating conditions, such as O₂ diffusion loss, temperature and relative humidity are all different in MEA and RDE.

For your convenience, we have highlighted the changes in the main text.

Reviewer #3 (Remarks to the Author):

Comments: *“Report for MS by Shan et al.*

The authors used an experimental and computational approach to study relationships between the coordination of surface atoms (shape of NPs) and dealloying-induced morphological changes of the Pt-Pd core shell structure. Although in situ TEM imaging is an important step in visualizing dealloying-induced transformations, this paper will only be suitable for publication in Nature Communications after a major revision. The authors must address the shortcomings listed below.”

Response: We thank the reviewer for encouraging us that in situ TEM is critical due to its capability in visualizing the materials structure transformations. However, the reviewer also commented on the shortcomings of our manuscript. Here we provide a point-to-point reply and revision to these suggestions, we think our manuscript is much improved after the revision.

Comments: *“The abstract is misleading. This paper has nothing to do with understanding how to minimize the corrosion of ORR catalysts. It is, in fact, a study related to monitoring morphological changes during the dealloying of Pt/Pd systems; and this should clearly be stated in the paper. The conclusion that we have to learn how to protect unstable, low coordinated atoms is not new and has been comprehensively discussed in the literature that, unfortunately, the authors omitted to cite”.*

Response: We thank the reviewer for the suggestion. In our new introduction, we have cited and reviewed the works on protecting the under-coordinated atoms on the surfaces using depositing or alloying with Au, adsorption of Br⁻, annealing and engineering the mesoporous structures (Line 51-54). These methods also help to improve the ORR activity of the catalysts.

Our work on the in situ observation of Pd@Pt nanocube corrosion shows the entire process of how the Pd core is etched from surface defects and corners, the analysis reveals the mechanisms including the galvanic etching and halogen etching. These findings are important in understanding the catalyst evolution during operation, which are also helpful to provide insights on how to optimize the catalyst structure for better activity and stability. We have modified our abstract to better fit with this focus now.

Comments: *“As mentioned above, it is important that it is now possible to use TEM for monitoring in situ morphological changes during dealloying of bimetallic systems. However, it is of paramount importance to use additional experimental probes that can provide atomic-scale information in order to be able to fully understand how the kinetics of dissolution depends on the density and nature of defects, the electrode potential, the electrochemical environment, and the history of the experiment. For example, in situ ICP-MS methods have been developed for studying single crystals, thin films and NPs, and in situ coherent X-ray diffraction has also been routinely used to study the role of defects in NPs. For details see: Angew. Chemie Int. Ed. 2012, 51 (50), 12613-12615; Electrochem. Commun. 48, 81-85 (2014); ACS Catalysis, 6 (2016) 2536-2544 and Sci-*

ence 356 (2017)739-742. Without these additional experimental techniques it is very easy to over-interpret the TEM results as, in my opinion, the authors have done in this manuscript.”

Response: We thank the reviewer for the suggestion. With extensive reading on the two in situ techniques using ICP-MS and grain Bragg coherent diffractive imaging (gBCDI), we now have better understanding on the strength and deficiency of different in situ approaches. In situ ICP-MS can measure the dissolution of metal electrodes in electrochemistry, while gBCDI can image the 3D strain and defect network in individual nanocrystals during different physical processes. However, in situ ICP-MS provides unique information on the average structure and composition, but the time resolution is only couple of seconds, which is beyond the time scale in the data from TEM with the time resolution of ~30 ms; gBCDI provide the morphology change of sub-200 nm individual grains, but local structure information including specific shape, localized etching, and the morphology and structure changes in the newly evolved areas can only be addressed by techniques with higher spatial resolution of ~1 nm in our case. Therefore we modified our introduction with more description on other important in situ techniques, and emphasized the advantage of in situ TEM study (Line 66-75). We also modified the conclusion (Line 289-297) to emphasize that the correlative study using various in situ approaches can offer more comprehensive understanding on the structure evolution and mechanisms. Thanks to the review's suggestion, in future we will definitely consider we will definitely consider combining these two techniques with in situ TEM when we study much slower materials evolution or chemical reactions at the scale of hundreds of nm.

Comments: “The authors have no proof for the proposed mechanism of Pt/Pd dealloying. In particular, there is no proof that Pd dissolution is accompanied with the ORR. Currently, it is well established that the dissolution of Pt-based metals in halide-free solutions is triggered by an irreversible, potential-induced oxide formation/reduction process. In the presence of halides, however, the dissolution is governed by the competition between metal-oxide formation and metal-halide complexation. It is surprising that the authors did not cite any work dealing with such issues. It is also surprising that the authors omitted to cite a classical work of the proposed dealloying mechanism for bimetallic systems (e.g., *Nature* 410, 450-453 (2001), *J. Am. Chem. Soc.* 134, 8633-8645 (2012), and *Nat. Mater.* 9, 904-7 (2010)).”

Response: We thank the reviewer for pointing out the necessity to have extra support on our proposed mechanisms. In the previous studies on the ORR activity of Pd@Pt nanoparticles, it has been clearly indicated that the Pd part was dissolved during the ORR test (Younan Xia, et al., *Nat Commun*, 2015, 6, 7594; Hui Zhang, et al., *Small*, 2016, 13, 1603423.). These works are now cited to support in our discussion to support the proposed mechanism (Line 218).

In our study, Br⁻ ions-induced dissolution is one type of the oxidative etchings, which is similar as the other type of oxidative etching by the sweeping electric potential in electrochemistry, which is helpful in understanding the corrosion process in bimetallic systems. Thermodynamically, the oxidation of Pd in the form of [PdBr₄]²⁻ could be easier than the route via the formation of oxides considering the redox potential difference between Equation (1) and Equation (3) in the Supplementary Section 2. Based on this

reason, in comparison with the competition between metal-oxide and metal-halide complexation formation, our study focuses on the interaction between halogen etching and galvanic etching. We also agree with the reviewer that discussion on the competition between metal-oxide formation and metal-halide complexation is needed. Now we have comprehensively compared these two species in the discussion part with the references suggested by the reviewer in Line 216-224.

Comments: “*The authors proposed that only Pd dissolution takes place during the dealloying; why not Pt as well? Certainly, the kinetics of dissolution are much higher for Pd than Pt, but still a small amount of Pt dissolution may have a substantial effect on the formation of the “hollow structure”. The authors must provide evidence that only Pd dissolution takes place during the dealloying.*”

Response: Thanks for the referee’s comment. We have considered the Pt dissolution accompanied by the Pd dissolution. According to the reduction potentials of $[\text{PdBr}_4]^{2-}$ and $[\text{PtBr}_4]^{2-}$ as follows:

when Pd and Pt are physically in contact in electrically conducting liquid, the Pd with lower redox potential serves as the anode while Pt serves as the cathode during the galvanic corrosion (*Principles and Prevention of Corrosion*, Denny A. Jones). The dissolution of Pd should be dominant before being consumed completely, however, a small amount of Pt dissolution into the form of $[\text{PtBr}_4]^{2-}$ may still take place. Pt atoms may also be lost due to the loss of the Pd support. We therefore do not exclude the possibility of Pt dissolution during the course of the process. The dissolution of Pt atoms on the surface of the shell could form new surface vacancies and defects that will further enhance the etching of Pd, due to the more accessibility of the Br^- ions to the Pd core. It is worth noticing that, in the later part of our experiment, after the Pd cubic core is etched away, the dissolution of resident Pt cage was slow, so that the corrosion behavior is barely observed (Movie S2-S4, in Figure S3 after 56 s, Figure S4 after 40 s, and Figure S5 after 50 s, respectively). We have corrected the corresponding description in the revised manuscript (see Line 224-228) and Supplementary Section 2.

Comments: “*The authors must also discuss the role of possible surface diffusion of the nobler element (Pt in this case) and how this may affect the final dealloyed structure.*”

Response: Thanks for the referee’s comment. We have read more related papers about the surface diffusion phenomenon in the process of corrosion and dealloying. Indeed, Pd and Pt have the potential of co-diffusion. Younan Xia, et al. reported the shell formation of Pt and Pd alloys after co-diffusion (Younan Xia, et al., *Science*, 2015, 349, 412. Younan Xia, et al., *Nano Lett.*, 2016, 16, 1467.). While atomic diffusion of Pt and Pd is difficult to detect in in situ liquid cell TEM, by comparing the ex situ characterization of EDS mapping on particles before and after etching (Figure S7 and Figure 3), we found some Pd species in the Pt shell, indicating that Pd-Pt alloy formed in the shell. This process is similar to the mechanism proposed by Heggen et al. (*J. Phys. Chem. C*, 2012, 116, 19073) for elements dissolution and rearrangement during electrochemical dealloying process of Pt-Co nanoparticles. We have added description about the surface diffusion of Pt atoms in the revised manuscript (Line 229-231). The Pt diffusion fa-

facilitates Pd atoms from inside to diffuse onto the Pt shell and form Pd-Pt alloy in the shell.

The final cage structure is mainly determined by the mechanic stability of the cage, which is beyond the topic of this paper. We do agree with the reviews that the collision of cage is interesting. We will continue studying this dynamic process by analyzing the surface and body strain change of cage during the process.

Comments: *“In conclusion, the paper has nothing to do with understanding how to make more durable catalysts for the ORR, and this should clearly be stated in the manuscript. Dealloying is a synthesis method that has been used for decades to make new porous architectures that may provide more active sites for the reaction of interest. A new direction should be to learn how such porous materials may affect the stability of surface atoms during electrocatalytic processes. In the revised version of the paper the authors must find a way to address correlations between their porous structure and the stability of this material.”*

Response: We thank the reviewer again for help to better deliver the true value to the community. In conclusion, we have also emphasized the importance of our works to help understand the corrosion mechanism and the structure evolution. The etching process, if under control, could also help to design more stable catalysts with porous or hollow architectures, which may provide more active sites in catalysis.

REVIEWERS' COMMENTS:

Reviewer #1 (Remarks to the Author):

This is an excellent work and the authors have done a thorough job of responding positively to the referees comments. I recommend accept as is.

Reviewer #3 (Remarks to the Author):

Report for MS by Shan et al.

The authors have mostly responded to my comments. The paper has been revised accordingly and it is now suitable to be published in Nature Communication.

Additional comments:

1. As stated before, I believe that in situ TEM imaging is an important step in visualizing dealloying-induced transformations, and this is the reason why this paper deserves to be published in Nature Communications.
2. I do not believe, however, that the shaped nanoparticles would play any major role in electrocatalysis of the ORR, as discussed by D. Li et. al., in *Energy Environ. Sci.*, 7(2014)4061-4069. The authors may consider citing this work simply because it may give a reader an opportunity to make his/her own conclusions about the role of particle shape on the ORR.
3. Finally, the authors may consider adding few sentences about recently published work by Yong-Tae Kim et al., in *Nature Communications*; DOI 10.1038/s41467-017-01734-7. In the paper, the authors discussed in details how new material porous architectures may lead towards unique activity-stability relationships. The authors may also consider going in the same direction.

Reviewer #1 (Remarks to the Author):

Comments: *“This is an excellent work and the authors have done a thorough job of responding positively to the referees comments. I recommend accept as is.”*

Response: We would like to thank the reviewer for the agreement to the publication of our work.

Reviewer #3 (Remarks to the Author):

Comments: *“Report for MS by Shan et al.*

The authors have mostly responded to my comments. The paper has been revised accordingly and it is now suitable to be published in Nature Communication.”

Response: We would like to thank the reviewer for the agreement to the publication of our work.

Comments: *“As stated before, I believe that in situ TEM imaging is an important step in visualizing dealloying-induced transformations, and this is the reason why this paper deserves to be published in Nature Communications.”*

Response: We thank the reviewer for appreciating the importance of in situ TEM. We truly believe by using this technique, more and more chemical reactions can be further deeply studied.

Comments: *“I do not believe, however, that the shaped nanoparticles would play any major role in electrocatalysis of the ORR, as discussed by D. Li et. al., in Energy Environ. Sci., 7(2014)4061-4069. The authors may consider citing this work simply because it may give a reader an opportunity to make his/her own conclusions about the role of particle shape on the ORR.”*

Response: We thank the reviewer for the suggestion. We have added this literature in the revised manuscript as a new ref. #11 in Line 44. We hope our work can offer readers a new perspective in the application and the durable electrocatalysts for ORR.

Comments: *“Finally, the authors may consider adding few sentences about recently published work by Yong-Tae Kim et al., in Nature Communications; DOI 10.1038/s41467-017-01734-7. In the paper, the authors discussed in details how new material porous architectures may lead towards unique activity-stability relationships. The authors may also consider going in the same direction.”*

Response: We thank the reviewer for the suggestion. We have added this literature in the revised manuscript as a new ref. #63 in Line 293. We hope our work can help to design porous materials with high performances in electrocatalysis in the future.